behaviour, ecology, theoretical biology

collective decision making, collective intelligence, consensus decision, voter model, wisdom of crowds

**Authors for correspondence:**
Claudia Winklmayr
e-mail: claudia.winklmayr@gmail.com
Albert B. Kao
e-mail: albert.kao@gmail.com

†These authors contributed equally to this study.

# The wisdom of stalemates: consensus and clustering as filtering mechanisms for improving collective accuracy

Claudia Winklmayr[1,2,†], Albert B. Kao[3,†], Joseph B. Bak-Coleman[4,5,6] and Pawel Romanczuk[1,7]

[1]Bernstein Center for Computational Neuroscience, Berlin, Germany
[2]Max Planck Institut für Mathematik in den Naturwissenschaften, Leipzig, Germany
[3]Santa Fe Institute, Santa Fe, NM, USA
[4]Department of Ecology and Evolutionary Biology, Princeton University, Princeton, NJ, USA
[5]Center for an Informed public, and [6]eScience Institute, University of Washington, Seattle, WA, USA
[7]Institute for Theoretical Biology, Department of Biology, Humboldt Universität zu Berlin, Germany

CW, 0000-0002-8784-2301; ABK, 0000-0001-8232-8365; JBB-C, 0000-0002-7590-3824

Groups of organisms, from bacteria to fish schools to human societies, depend on their ability to make accurate decisions in an uncertain world. Most models of collective decision-making assume that groups reach a consensus during a decision-making bout, often through simple majority rule. In many natural and sociological systems, however, groups may fail to reach consensus, resulting in stalemates. Here, we build on opinion dynamics and collective wisdom models to examine how stalemates may affect the wisdom of crowds. For simple environments, where individuals have access to independent sources of information, we find that stalemates improve collective accuracy by selectively filtering out incorrect decisions (an effect we call stalemate filtering). In complex environments, where individuals have access to both shared and independent information, this effect is even more pronounced, restoring the wisdom of crowds in regions of parameter space where large groups perform poorly when making decisions using majority rule. We identify network properties that tune the system between consensus and accuracy, providing mechanisms by which animals, or evolution, could dynamically adjust the collective decision-making process in response to the reward structure of the possible outcomes. Overall, these results highlight the adaptive potential of stalemate filtering for improving the decision-making abilities of group-living animals.

## 1. Introduction

Collective decision-making is an essential feature for organisms across a wide range of taxa, from bacteria to fish to humans [1]. For some species, individuals accrue benefits from group living for reasons unrelated to the decision-making process and make consensus decisions simply to maintain cohesiveness [2,3]. Beyond cohesion, many other species make decisions collectively in ways that improve accuracy and the fitness of the animals within the group [4,5].

The potential for decision accuracy to increase with group size is often referred to as the 'wisdom of crowds', 'collective wisdom' or 'collective intelligence'. Traditional models assume that the increase in accuracy occurs because individuals contribute different, and somewhat uncorrelated, pieces of information relevant to the decision. Because the group as a whole has access to a greater amount of information than any single individual, the resulting collective decision has the potential to be more accurate than is possible for an individual. A wide variety of theoretical models, including the well-known Condorcet jury theorem [6], but

also much recent work loosening some of the theorem's assumptions (e.g. [7–9]), and an increasing number of empirical studies (e.g. [10,11]), have indeed demonstrated this effect for different contexts and species.

Models of collective wisdom often explicitly assume decision-making processes inherently lead to consensus, typically as a result of a voting process such as majority rule, quorums [9,12,13], or an averaging of individual opinions [14]. Realistic decision-making in animal groups, however, is not exogenously aggregated and instead relies on endogenous processes of typically local social interactions. For example, in fish schools and bird flocks, the trajectories of individual animals are influenced by both the movements of their near neighbours, as well as their own preferences, and these myriad momentary interactions may result in coherent collective movement towards a single direction of motion [15–18]. While these dynamics may at times approximate majority rule [12,13,19], the mapping between the microscopic social interactions and macroscopic collective decisions remains an active area of research. In addition to the social interactions of group-living animals such as fish and birds, the endogenous decision-making capability of other collective systems have also been modelled, such as neural systems [20] and insect colonies [21,22], where different mechanisms may lead to effective, and sometimes optimal, collective decision-making.

Crucially, emergent decision-making processes through social interactions may not guarantee that a group reaches a consensus within a reasonable time period, or at all. Therefore indecision or a stalemate can be considered an effective third option whenever a group is faced with a binary decision problem. Whether this option is in fact viable will then depend on the relative cost of a stalemate compared with a wrong decision, which will be strongly context dependent.

For example, many small schooling fish are prey to larger predators and prefer to hide in grasses or shelters for safety [23]. Before leaving the shelter they will have to decide where to travel (e.g. to one out of two or more possible food sources). These decisions may depend on individually sensed information about foraging opportunities and the likelihood of encountering a predator. Failure to reach a consensus about the destination of travel, and consequently staying in the shelter, may be a relatively low-cost option (compared with being eaten). For simplicity, we will assume initially that costs of stalemates are negligible and focus on how they affect collective accuracy (i.e. the probability that the group selects the better of the two non-stalemate options). Later on, we will discuss the role of stalemate costs with a detailed discussion in the electronic supplementary material.

Furthermore, recent work has suggested that the specific informational environment in which collective decisions are made can have major effects on the resulting decision accuracy. In particular, a combination of both uncorrelated and correlated information can interact, resulting in low decision accuracy for large group sizes, with decision accuracy instead being maximized by intermediate-sized groups [19,24,25]. This stands in contrast to the predictions of many wisdom of crowds models, including the Condorcet jury theorem, which predict a monotonic increase in collective accuracy as group size increases.

The presence of both uncorrelated and correlated information may frequently occur in nature due to differences in the degree of spatial and temporal correlation of different environmental cues. For example, loud auditory cues may be highly correlated across individuals in a group, while visual cues may be more localized and therefore less correlated across individuals due to the limited visual field of each animal, and occlusion of one's field of view by the bodies of neighbours. Another way in which uncorrelated and correlated information can be present simultaneously is from social copying within a group. A theoretical model has demonstrated that if individuals begin with independent (uncorrelated) information, but make individual decisions by incorporating the previous decisions of other group mates, then a similar phenomenon of an optimal intermediate group size, and poor decision accuracy for large groups, can also emerge [26].

Opinion dynamics models provide a flexible method to examine the mapping between social interactions, environmental cues, collective decisions, and decision accuracy [27]. In such models, each individual begins with some initial opinion (e.g. determined by the environmental cues that the individual observes), and then the opinions may change in time depending on the social network structure and the social interaction rules that individuals follow. These dynamics may lead to consensus, whereby all individuals have the same opinion, or may fail to reach consensus, by arriving at some other state where the system indefinitely retains a mixture of opinions.

In order to better understand how animals may make collective decisions in naturalistic conditions, here we examine the effect of opinion dynamics on collective decisions in both simple environments (i.e. individuals independently sample a single environmental cue, identical to the context described by the Condorcet jury theorem) and complex environments (i.e. individuals can sample from both a correlated and an uncorrelated environmental cue). While the decision scenarios that we highlight here (particularly the Condorcet jury theorem) are overly simplistic compared with the actual situations that many real animal groups face, they provide a useful starting point to examine how opinion dynamics interact with collective wisdom. As we show, even such relatively simple environments are sufficient in highlighting interesting phenomena, particularly that opinion dynamics can substantially alter the outcome of collective decisions, and in most cases, improve collective accuracy compared with simple majority rule.

## 2. Results

### (a) Stalemates can improve wisdom of crowds in simple and complex environments

If a group, faced with the decision between two options (e.g. two potential food patches, fleeing directions, locations to rest) fails to come to a consensus, it effectively chooses a third option: to do nothing. Throughout this work, we will assume that groups are faced with binary decisions (i.e. choices between two options, where one option is 'correct' and the other is 'incorrect'). Because here we assume that the cost of stalemates is negligible, we do not consider their contribution to a group's accuracy. Decision accuracy is then calculated as the number of trials where a correct consensus decision was made divided by the total number of trials that reached consensus.

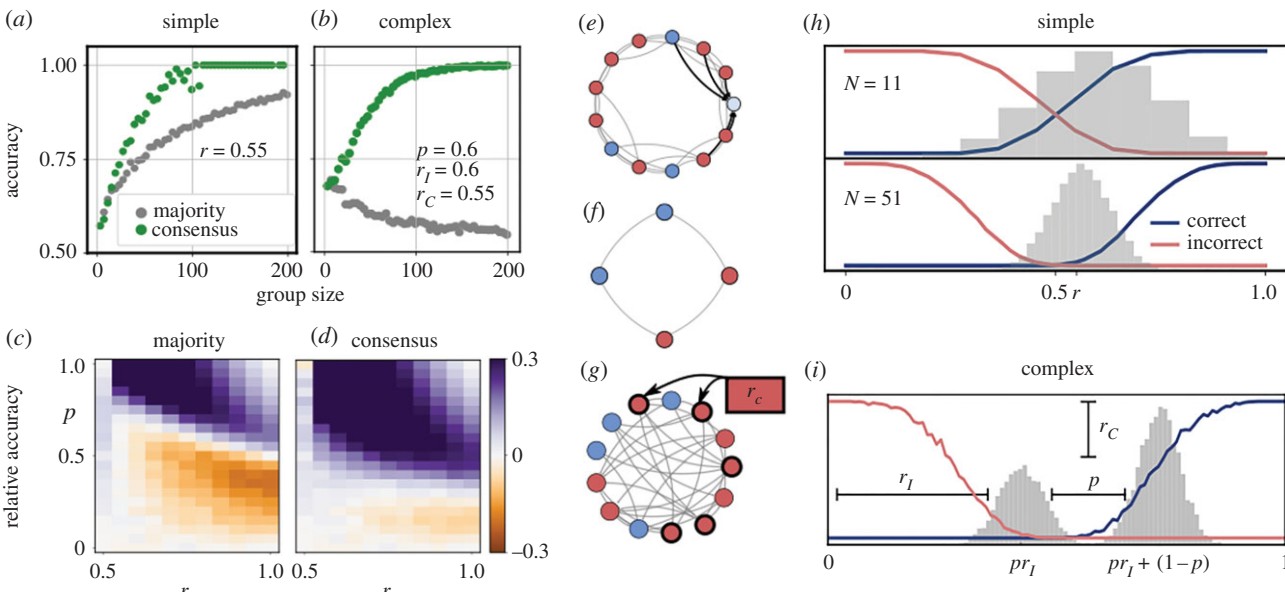

**Figure 1.** Consensus decision making and stalemate filtering in simple and complex environments. (*a*) In simple environments with low cue reliability, groups which reach consensus through opinion dynamics (green) show higher collective accuracy than groups which use simple majority vote (grey). Both decision making procedures show an increase in accuracy with group size (i.e. wisdom of crowds). (*b*) When majority voting is employed in complex environments the wisdom of crowds can break down such that an increase in group size can lead to a decrease in collective accuracy (grey). Using opinion dynamics for consensus formation can remedy this effect and restore the wisdom of crowds (green). (*c*) Regions in the $r_I \times p$ parameter space where groups of $N = 51$ individuals using majority vote perform better (purple) or worse (orange) than a solitary individual. $r_C$ is set to 0.55. (*d*) When groups use opinion dynamics to reach consensus the region in parameter space where groups outperform individuals increases. All parameters are the same as in (*c*). (*e*) Example of a single updating step in a highly clustered (WS) network ($\beta = 0.1$). The focal node (light blue) observes the opinions of $k = 5$ immediate neighbours and will change its opinion from blue to red. (*f*) A minimal example of a network that has reached a stalemate. Because each node has one blue and one red neighbour, none of the nodes will change their opinion. (*g*) Example of the formation of initial opinions in a complex environment. With probability $1 - p$, an individual attends to the correlated source (red box), and all individuals that follow this source will receive the same information (here red). All other agents sample independently observed information which is correct with probability $r_I$. (*h*) Probability that a group reaches a consensus for the correct (blue) or incorrect (red) option as a function of the initial fraction of individuals voting for the correct option for small ($N = 11$) or intermediate ($N = 51$) group sizes. The grey histograms illustrate the distribution of initial votes for a cue with reliability $r = 0.55$. As group size increases, the initial vote distribution needs to be increasingly biased in order for a consensus to be reached (i.e. the inflection point of the red and blue curves shift to more extreme values). Assuming that the cue is informative (i.e. $r > 0.5$), the set of initial opinions will tend to have a positive bias, and the opinion dynamics will tend to reach consensus towards the correct option. (*i*) In complex environments, the distribution of initial votes is bimodal. The centres of the modes correspond to the conditional probability of an individual being correct, given that the correlated cue is correct (right mode) or incorrect (left mode). Black lines illustrate the effect of the three model parameters on the shape of the distribution: $r_I$ determines the distance from 0, $r_C$ determines the relative heights of the two modes, and $p$ governs the distance between the modes. Red and blue lines depict the probability of a correct or incorrect consensus as in (*h*). Consensus is unlikely when the correlated cue is incorrect (left mode). (Online version in colour.)

In our model, individuals in a group are connected to one another into a social network, represented by a graph where the nodes denote the individuals and connections between individuals are shown as edges. For the main part of this work we will focus on Watts–Strogatz (WS) networks, a well-studied family of networks that spans the range from highly clustered to random networks [28,29]. We were also able to reproduce our results on scale-free networks and real biological networks formed by fish schools (for a more detailed discussion see electronic supplementary material, sections S3–S4. WS networks are particularly convenient because they are completely determined by only three parameters: the group size ($N$), the node degree (i.e. the number of neighbours that an individual is connected to, $k$) and the rewiring probability ($\beta$). The rewiring probability specifies the probability, when constructing the network, that a node is connected to a randomly selected node instead of a spatially close node, thus controlling the extent to which the network is highly clustered ($\beta = 0$) or random ($\beta = 1$). Once a network has been generated, it remains static for the duration of the decision-making process. We examine groups ranging in size from $N = 3$ to 200, and, for simplicity, initially fix the number

of neighbours to $k = 5$ (and a fully connected network if $N < k + 1$) for each individual (which approximates the number of neighbours that animals tend to pay attention to when making collective decisions [30]) and the rewiring probability to $\beta = 0.2$. However, we later explicitly examine the role that $k$ and $\beta$ play in affecting collective accuracy, as well as our model's extensibility to other types of graphs.

Next, the social interaction rules must be specified, which govern how individuals change their opinion due to the opinions of others. To approximate the social interactions among real animals, we assumed an asynchronous updating policy where at each time step, a random focal individual is selected, and that individual changes its opinion to the majority opinion of its neighbours (figure 1*e*; in the case of a tie among the neighbours, the individual's opinion is left unchanged, figure 1*f*) [31,32]. Such 'threshold' or 'majority' voter models dynamics are not guaranteed to reach consensus; instead, the network may become 'stuck' in an intermediate state or oscillate between intermediate states (a detailed discussion of the convergence behaviour of the threshold model can be found in [33], and a minimal example of a stuck network is shown in figure 1*f*). In order to detect whether a

given network has reached a stalemate we probe the simulation every $n$ update steps and check if any node still holds an opinion different from the majority of its neighbours (i.e would switch to the opposite opinion when selected for updating). If this is not the case the network has either come to consensus or reached a stalemate and the trial is over. The value of $n$ is determined as a function of group size, for more details see electronic supplementary material, section S1.

To set the initial opinions of each individual (pre-interaction) we distinguish between informationally simple and informationally complex environments. In simple environments, we assume, as in the Condorcet jury theorem, that each individual has access to an independently sampled environmental cue. As a result, the initial opinions are independent of each other, and each has a probability $r$ of indicating the correct option, called the cue's 'reliability'. We assume that $r > 0.5$ (i.e. that the cue is positively informative, otherwise individuals could simply reverse their interpretation of the cue to generate an informative cue). As is well-known from the Condorcet jury theorem and related work, if $r > 0.5$, and a group makes decisions using simple majority rule then the probability that the group makes a correct decision increases monotonically with group size and asymptotes at perfect accuracy (the so-called 'wisdom of crowds') [14].

In informationally complex (and more naturalistic) environments, we follow previous studies [19,24,25] in assuming that there are two cues in the environment. One cue (the uncorrelated cue) is independently sampled by each individual in the group, and has reliability $r_I > 0.5$. The other cue has reliability $r_C > 0.5$ and is correlated across all of the individuals in the group, such that all of the individuals observe the same information from that cue in any given trial. Whether an individual attends to the independent or the correlated cue, is governed by the 'voting strategy' $p$. With probability $p$ an individual's initial vote is a sample of the independent cue and with probability $1 - p$ the vote is the current value of the correlated cue.

Figure 1 provides an overview of the effects of consensus decision making on collective accuracy. First of all, the examples in figure 1a,b show that in both simple and complex environments consensus decision making via opinion dynamics leads to a substantial increase in collective accuracy compared with using simple majority rule, where initial votes are averaged without any opinion exchange. The effect is particularly strong for complex environments. Previous work [24] has shown that if $p < 1/(2r_I)$ (as is the case in figure 1b), groups using simple majority will not display wisdom of crowds, but rather achieve maximum accuracy at finite group sizes. Very large groups will asymptote at an accuracy of just $r_C$, since in this region of parameter space the correlated cue tends to dominate the collective decision. When opinion dynamics are used in this region of parameter space, we find that the wisdom of crowds can be restored, and the dominance of the correlated cue negated, with collective accuracy again increasing monotonically with group size and asymptoting at perfect accuracy.

To illustrate the effect of consensus decisions in complex environments more broadly, we performed a parameter scan across the entire $r_I \times p$ plane, while keeping the reliability of the correlated cue fixed at $r_C = 0.55$ (figure 1c,d). The two panels show the relative collective accuracy of a group of size $N = 51$ compared with the accuracy of a solitary

individual. We find that the region in parameter space where groups perform better than solitary decision makers is larger when opinion dynamics ($d$) are used than when majority rule is employed ($c$).

Figure 1h,i illustrates the mechanism underlying the above observations: the probability of a correct (incorrect) consensus decision, as indicated by the blue (red) lines, increases nonlinearly with the proportion of correct (incorrect) initial opinions (the probabilities of correct and incorrect consensus do not necessarily sum to one because of the non-zero probability of stalemates). In particular, when the proportion of initial correct opinions is $\approx 0.5$ groups, especially large ones, are highly unlikely to reach consensus. Consensus reliably occurs only when the initial opinions are highly biased.

This explains the ability of stalemates to act as a filter to improve collective decisions (hence the term *stalemate filtering*). In simple environments the probability that a certain proportion of initial opinions is correct, follows a binomial distribution with mean $r$ (figure 1h, grey histograms). Assuming a positively informative cue ($r > 0.5$), strong bias and thus consensus is more likely to occur for the correct option. Stalemates therefore indirectly reject scenarios where the majority opinion is wrong, and boost the probability of correct decisions. If $r < 0.5$, the scenario would, of course, be reversed, i.e stalemates would then filter out trials where there is an initial majority for the correct option. But in this case individuals could just reverse the interpretation of the cue as discussed above.

In complex environments, we observe a bimodal distribution of initial opinions, one mode resulting from cases where the correlated cue is correct, and the other mode resulting from cases where the correlated cue is incorrect (figure 1i, grey histograms). We can demonstrate that the initial opinions are, on average, less biased when the correlated cue is incorrect, compared with when the correlated cue is correct: the left mode is closer to 0.5 than the right mode if: $(pr_I - 0.5)^2 < ((pr_I + (1 - p)) - 0.5)^2$. This inequality holds if $r_I > 0.5$, which is what we assume in our model.

Because of this, stalemates are more likely to occur when the correlated cue is incorrect, while consensus is more likely to occur when the correlated cue is correct. Therefore, stalemates effectively reject the correlated cue when it is incorrect, which serves to break the dominance of the correlated cue and restore the wisdom of crowds in complex environments. Notably, as presented in electronic supplementary material, section S5, we demonstrate that stalemate filtering can further improve collective decision accuracy even if groups employ an optimal voting strategy $p$ as introduced in [19].

## (b) Spatial clustering and sparseness improves the filtering of inaccurate decisions

In general, we find that the higher the probability of a stalemate, the more accurate the collective decision if consensus is reached, across all parameter space, in both simple and complex environments. Therefore, to understand how different properties of the network affect collective accuracy, we need only to examine how these properties affect the probability of stalemates.

We examine the role of four network properties on the probability of stalemates: the group size ($N$), the rewiring probability ($\beta$), the node degree $k$ (here expressed as the normalized degree, $k/N$), and the randomness of the distribution of initial opinions in the network (where a value of 0 indicates that like opinions are maximally clustered in the

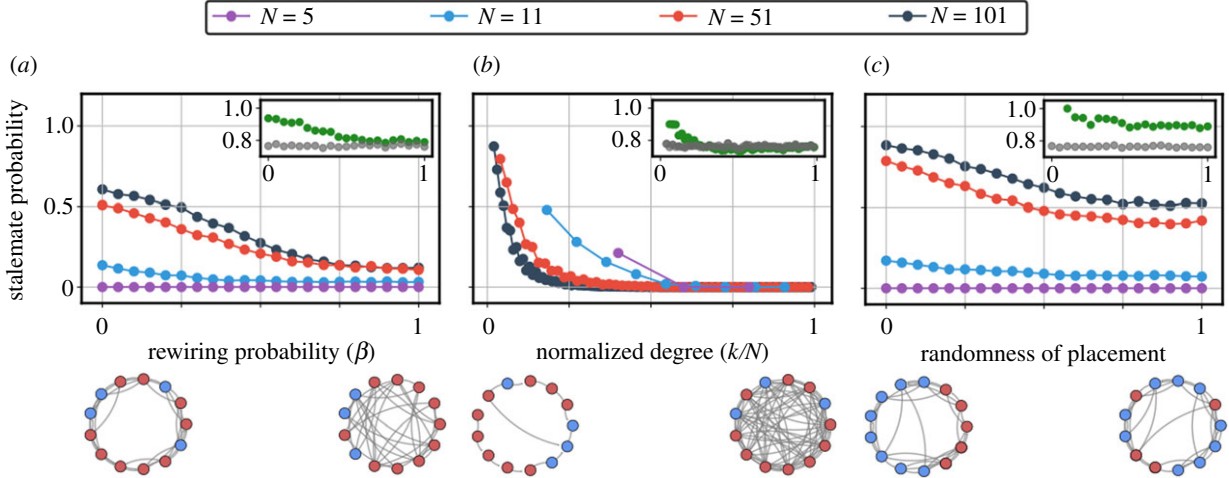

**Figure 2.** The effect of network structure on the probability of stalemates. These results are averages over all possible initial opinion configurations. We generally assume a default of $\beta = 0.2$, $k = 5$ and initial opinions that are randomly distributed within the network. Each panel shows the effect of varying a particular parameter while keeping the other two fixed. The insets show the effect of the respective structural parameters on collective accuracy for a group of $N = 51$ individuals in a simple environment with a cue of reliability $r = 0.55$. Green dots show collective accuracy at consensus and grey dots the result of a majority vote (unaffected by the structural parameters). In all three cases we find high values of collective accuracy to be linked to high probability of stalemate. (a) The probability of stalemates increases as the rewiring probability shrinks (i.e. the network becomes more clustered). This is particularly true for larger networks. (b) The probability of stalemates increases as the average number of neighbours that an individual is connected to decreases (i.e. the network becomes sparser). (c) The probability of stalemates increases as the distribution of initial opinions becomes more clustered. When the randomness parameter is 0, all nodes with the same opinion are placed next to each other in the network, and when the parameter is 1, initial opinions are randomly placed on the network. (Online version in colour.)

network, and a value of 1 indicates that initial opinions are randomly distributed in the network). We find that the probability of stalemates tends to increase as group size increases, as the interaction network becomes more clustered, as the number of neighbours decreases, and as the initial opinions are increasingly clustered (figure 2).

Those networks where stalemates achieve the strongest filtering effects are those that are large, highly clustered (with respect to network structure and distribution of initial opinions), and sparse. Such social structures may in fact be common in nature, where visual occlusion [34,35] or other mechanisms generate clustering in animal groups [25]. Furthermore, existing empirical evidence suggests that, for many social animal species, individuals pay attention to their closest 1–7 nearest neighbours [17,18,30,36], resulting in relatively sparse social networks.

In addition to studying the role of structural parameters in WS graphs we also investigated the effect of global properties of the interaction dynamics, namely decision time and other types of graphs (see electronic supplementary material, sections S2–S4). In particular, we found that if decision time is limited (i.e. the decision process is terminated after a fixed number of updating steps), collective accuracy increases. This ties into the previous findings in that limiting the decision time will make consensus harder to achieve and thus only initially biased groups (which will generally be biased towards the correct option) are able to reach a decision quickly.

Also the type of network connecting the individuals affects the consensus probability and thus the collective accuracy. In scale-free networks, such as those generated with the Barabàsi–Albert model, groups are highly likely to reach consensus (except in cases of extreme sparseness) and thus benefit little from stalemate filtering.

We also tested our findings in biological networks, inferred from position data of schooling fish, before and after the application of Schreckstoff, a natural alarm substance that increases school cohesion leading to an increased normalized degree. Here we could confirm that networks with lower normalized

degree were more likely to end up in a stalemate but also showed the highest collective accuracy at consensus. This might indicate that an increase in cohesion as a response to external threats might not only serve physical protection but also facilitate fast consensus decisions in situations where indecision might be fatal. For more details on these results, refer to the electronic supplementary material, section S4.

These findings have two main consequences. Firstly, knowing the network structure and the specific decision context, it is possible to predict the likelihood that a group reaches consensus through opinion dynamics. Secondly, individuals in the group may modify the network structure through changes in their social interaction behaviour in order to optimize the trade-off between consensus probability and decision accuracy.

## (c) Detecting and breaking stalemates

So far we have shown that in settings where decision processes are likely to end in a stalemate, whenever a consensus is reached it is likely to be a consensus for the correct option. It is, however, not obvious how the group members would be able to determine whether consensus or a stalemate has been reached on a global level. For example, individuals within a group might believe that full consensus was achieved and act accordingly, although a certain (small) fraction of individuals remains unconvinced of the majority opinion. We therefore tested the robustness of our findings with respect to the potential misjudgement of the global decision state. In figure 3, we lower the effective consensus threshold from $q = 1$ to $q \in \{0.95, 0.9, 0.8\}$. Figure 3a,b shows that the general mechanism of stalemate filtering is robust to such a change. Although the achieved accuracy boost is lower than for full consensus, collective accuracy still increases with group size and in all cases lies considerably above the values achieved with simple majority voting.

While stalemates indirectly help to improve collective accuracy by filtering, groups can only reap these benefits

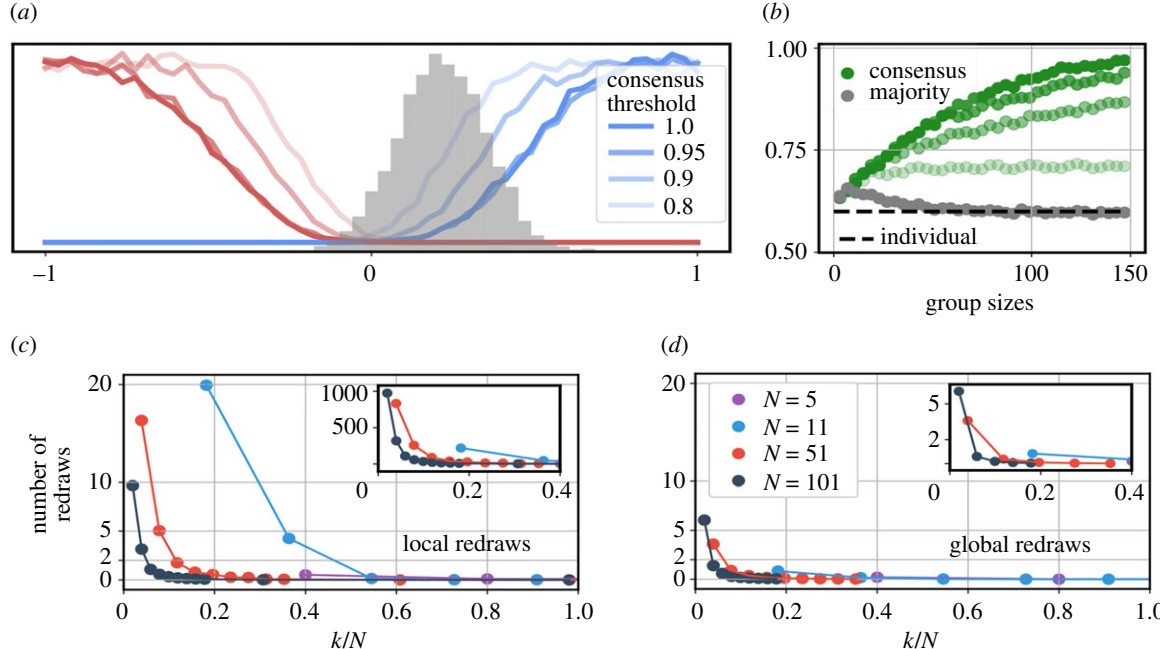

**Figure 3.** Detecting and breaking stalemates. (*a*) A reproduction of figure 1*h* with $N = 51$ and different levels of consensus (80, 90, 95 and 100%). As the consensus threshold is lowered, the fraction of trials reaching consensus increases, however the general shape of the consensus curve remains the same. (*b*) Reproduction of figure 1*a*, *b* for different levels of consensus colour intensities correspond to the legend in (*a*). (*c*) Average number of local redraws per individual needed for consensus to be reached as a function of the normalized degree $k/N$. Low values of $k/N$ are associated with a high stalemate probability (see figure 2) and thus require more redraws per capita than densely connected networks. The inset shows the absolute number of redraws (i.e not normalized by group size) for very sparsely connected networks. (*d*) Average number of global redraws needed for consensus to be reached. Both *c* and *d* show results for networks of different sizes, the colour coding is the same as in figure 2. (Online version in colour.)

once a decision for one of the options is reached. This means, groups must generally aim to break the stalemate. One strategy would be to change the relevant network parameters, but while this does increase the consensus probability it would simultaneously diminish the benefits of stalemate filtering. Another, more plausible strategy would be for a group to interpret the inability to reach consensus as an indication that the initial information was insufficient, leading individuals to seek out more external information. This resampling procedure could either be performed on a global or a local scale.

Local resampling means that if an individual observes a stalemate among its neighbours (which does not generally imply a global stalemate) this individual will reach out for new information from the environment. This new information is drawn from the original distribution (i.e. keeping $p$ and $r_C$ fixed and using the same value of the correlated cue). Figure 3*c* shows how many local redraws per individual are needed (on average) to resolve a stalemate. Global resampling means that if the whole group gets stuck in a stalemate every individual will redraw new information using the same distribution as initially. This is equivalent running a new trial. Figure 3*d* shows (for different group sizes) the average number of full redraws needed until consensus reached as a function of the normalized degree ($k/N$). If the probability of stalemates is high (e.g. in the case of large groups or sparsely connected networks) groups will need more redraws on average. However, in most cases two or less redraws are enough to turn a stalemate into consensus.

## 3. Discussion

Our results highlight the importance of considering social decision-making dynamics that do not impose consensus

and cannot be summarized simply as a form of 'majority rule'. In relaxing this assumption, we observe that such dynamics can often result in a stalemate, whereby the group is unable to reach a consensus. In our model, stalemates tend to filter outcomes where the majority decision would be incorrect, both in simple and complex environments, resulting in an improvement in collective accuracy.

Individual animals in a group may, in principle, be able to titrate the frequency of stalemates by changing the proportion of long-distance connections, or the number of neighbours to which they're attending. In principle, such changes to network structure can be achieved by even minor changes to individual behavioural rules. For example, fish schools and other social animal groups have long been known to move closer together in response to increased predation risk. A dominant early explanation for this tendency was that the close proximity resulted from competition for low-risk places within the group [2]. More recent work has demonstrated, however, that moving closer alters the network of social interactions in a way that increases collective responsiveness [37–39]. Our analysis of networks of real fish schools demonstrated that the experimentally observed increase in cohesion as a response to a higher perceived risk results in an increased likelihood of consensus, but decreased decision accuracy (see electronic supplementary material, section S4). This appears to be a reasonable collective response in the given experimental context of a fish school in danger without any shelter [37], where any decision (e.g. fleeing to the left or the right) might be preferable to doing nothing.

This last point ties into a more profound question of the nature of decision problems. While many existing models (the present work included) focus mainly on decision accuracy (i.e. ask whether a given choice is right or wrong), a growing

body of literature emphasizes the value of the choices (e.g. quality of nesting sites or food patches) as the main driving factor of decision making [40]. Of particular interest, empirical evidence has shown that humans and other primates decide more quickly when shown two equally high-quality options compared with two equally low-quality options [41,42], with theoretical models replicating the 'value-sensitive hypothesis' [22,42]. Our results suggest that animals in groups could additionally adjust the structure of their social network in order to titrate between speed and accuracy of decision making. While a full analysis of our model in terms of value-based decision problems would go beyond the scope of the present work, we analysed a minimal reward framework, varying the cost of stalemates (electronic supplementary material, section S6). We find that a non-zero frequency of stalemates can be profitable even when there is a cost to stalemates. While these results are still preliminary, they indicate an interesting connection to value and reward-based models which future research should investigate further. For animal groups, using stalemates to filter out incorrect decisions would improve the fitness of the individuals in the group if the cost of indecision is low, relative to the cost of an incorrect decision, but not necessarily zero. This is likely to be the case across a broad range of behaviourally and ecologically relevant contexts. Staying put may be preferable to misjudging the presence of a predator, getting lost, or moving to a lower-quality foraging patch. In particular, stalemates can cause a group to gather more information before making a decision. This can allow a group to avoid costly mistakes and boost collective accuracy, and occasional stalemates may be evolutionarily favoured if the cost of indecision is relatively low to individuals in the group. However, for other species or contexts, the cost of indecision is not low. For example, if a predator is attacking the group, when a food patch becomes completely depleted, or when a shelter becomes uninhabitable, indecision may be costly, and making even a wrong decision may be preferable to a stalemate. Therefore, the particular cost-reward structure of the options available to an animal group may either incentivize, or deincentivize, groups to employ stalemates as part of their decision-making process. We note that there may be multiple mechanisms by which groups could create deadlocks in the decision-making process [22].

Our model may also be extended to investigate the effect of opinion dynamics, and stalemates, in other contexts. For example, there may be variation in the estimation ability among the individuals in a group, whether due to different prior experiences or abilities, or due to different perceptual abilities as a function of an individual's physical position within the group. Model extensions could examine whether more knowledgeable individuals could position themselves within the network structure to more strongly influence the collective decision [43], and more generally, how network structure [44] or social information sharing [45,46] can affect collective wisdom.

In summary, opinion dynamics within social groups can strongly affect the quality of collective decisions, and generating a quantitative mapping between the microscopic interactions between individuals and the resulting collective decision is crucial to an understanding of decision making for many animals groups and other collective systems [47]. In particular, the possibility of stalemates as an outcome of such dynamics is an understudied but potentially functionally important feature of collective decisions. By titrating the rate of stalemates, animals in groups may access an additional independent mechanism that they may exploit to improve the accuracy of collective decisions.

**Data accessibility.** The python code used for simulations can be found at https://github.com/ClaudiaWinklmayr/WisdomOfStalemates. An early version of this publication is available as preprint on *bioRxiv* [48].

**Authors' contributions.** P.R. and J.B.B.-C. conceived the study. C.W. and P.R. designed the study with the help of A.B.K. and J.B.B.-C. C.W. implemented the model and performed numerical simulations and mathematical analyses. C.W. analysed the simulation data with the help of P.R., A.B.K. and J.B.B.-C. A.B.K. wrote the paper with help from all authors. P.R., C.W. and J.B.B.-C. critically revised the manuscript. All authors gave final approval for publication and agree to be held accountable for the work performed therein.

**Competing interests.** We declare we have no competing interests.

**Funding.** C.W., J.B.B.-C. and P.R. acknowledge the support of the Cooperation and Collective Cognition Network (CoCCoN) funded by the strategic partnership between Princeton University and Humboldt Universität zu Berlin. A.B.K. acknowledges support from a Baird Scholarship and an Omidyar Fellowship from the Santa Fe Institute. P.R. acknowledges funding by the Deutsche Forschungsgemeinschaft (DFG, German Research Foundation) under Germany's Excellence Strategy—EXC 2002/1 'Science of Intelligence'—project number 390523135, as well as through the Emmy Noether programme, project number RO4766/2-1.

**Acknowledgements.** We gratefully acknowledge Matt G. Sosna and Iain Couzin (MPI for Animal Behavior, Konstanz) for providing us with fish position data.

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
