## [Reviewer comments · Proceedings of the Royal Society B: Biological Sciences]

Review History

RSPB-2020-0096.R0 (Original submission)

Review form: Reviewer 1

Recommendation

Accept with minor revision (please list in comments)

Scientific importance: Is the manuscript an original and important contribution to its field?

Good

General interest: Is the paper of sufficient general interest?

Good

Quality of the paper: Is the overall quality of the paper suitable?

Good

Is the length of the paper justified?

Yes

Should the paper be seen by a specialist statistical reviewer?

No

Do you have any concerns about statistical analyses in this paper? If so, please specify them explicitly in your report.

No

It is a condition of publication that authors make their supporting data, code and materials available - either as supplementary material or hosted in an external repository. Please rate, if applicable, the supporting data on the following criteria.

Is it accessible?

Yes

Is it clear?

Yes

Is it adequate?

Yes

Do you have any ethical concerns with this paper?

No

Comments to the Author

In this study the authors use a modelling approach to explore the potential influence of 'stalemates' on the process of collective decision making. They investigate how consensus decisions based on opinion dynamics perform relative to a simple majority rule in both 'simple' and 'complex' environments and then explore the role of network structure and rewards structure on these patterns.

I found the analysis to be thorough and interesting, and the presentation a rigorous analysis of the concept they present. The idea that stalemates should be considered as an important component of the decision process is certainly worthy of further investigation, and the treatment of network structure effects is revealing. The manuscript is in general well written, though could possibly be improved with some clarification of the modelling approach and results in places.

I do have some concerns with the biological applicability of the model and assumptions.

My greatest concern is with the fundamental premise that stalemates are not costly. I would argue instead that stalemates are, by definition, *always* costly. If an organism is required to make a decision, this implies that the current situation is not suitable and therefore that not deciding will induce a cost. As the decision process (particularly a collective one) is costly in itself, it makes no sense to initiate this process unless there is a cost of not doing so. A recent model of ant decision making during emigrations, for example, assumed that stalemates were equivalent to failure (Cronin 2019 J. Theor. Biol). I agree that the cost of a wrong decision may be higher than the cost of a non-decision, but this is likely why we observe adjustments of quorum thresholds in collective decisions where this the case, to insure against this outcome (eg Dornhaus et al 2004 Animal Behaviour).

The authors do make some effort to acknowledge the effect of reward structure in the final part of the manuscript and show that this can strongly influence the patterns observed. I wonder, however, how they can justify their general and strongly stated claims that stalemates improve decision making. In fact, it seems that this is true for only some contexts, and possibly unusual ones at that. I do not think that this detracts from the validity of the results in general, but I do think the authors need to be more cautious and transparent in explaining the context and limitations of their findings.

Minor comments:

I found it unclear of how the correlated information was applied to other individuals in the group. Perhaps this can be made clearer.

How does the 'simple majority' rule work in this simulation? I guess this is simply based on the proportion of initially correct opinions without any iteration?

Opinion dynamics seems to augment the Presumably the opinion polling method would also lead to lower success for lower values of r (<0.5) and that stalemates would then filter out good choices? I think it is worth mentioning this 'dark side' of this method of decision making also relative to the simple majority rule if this is true.

In figure 4, the probability of stalemates appears to have not reached an equilibrium for larger groups, so are we really seeing a 'stalemate' or is this simply an effect of the greater complexity of consensus building in larger groups? This brings into question the definition of a stalemate. Similarly, it is stated that S_{max} is set to "twice the average number of steps for consensus". Was this 'for each group size'? This should be made clear as the average is clearly very different depending on what is averaged.

Review form: Reviewer 2 (James Marshall)

Recommendation

Major revision is needed (please make suggestions in comments)

Scientific importance: Is the manuscript an original and important contribution to its field?

Acceptable

General interest: Is the paper of sufficient general interest?

Acceptable

Quality of the paper: Is the overall quality of the paper suitable?

Acceptable

Is the length of the paper justified?

Yes

Should the paper be seen by a specialist statistical reviewer?

No

Do you have any concerns about statistical analyses in this paper? If so, please specify them explicitly in your report.

No

It is a condition of publication that authors make their supporting data, code and materials available - either as supplementary material or hosted in an external repository. Please rate, if applicable, the supporting data on the following criteria.

Is it accessible?

Yes

Is it clear?

Yes

Is it adequate?

Yes

Do you have any ethical concerns with this paper?

No

Comments to the Author

The authors show that filtering decisions that are likely to be incorrect increases accuracy on the remaining collective decisions; in itself this is obvious, but interestingly the authors use non-convergence of opinion dynamics on networks to achieve the filtering. However I have serious concerns about the biological relevance of the work.

The general question, of how to achieve decisions on networks where there is no centralised tallying of votes is interesting; however, this manuscript begs further questions for me... in particular, how is consensus or absence thereof to be assessed by group members without precisely such a centralised overseer? I suppose one could assume that group members will monitor their neighbourhood and after a prolonged period of observing no changes of opinion conclude that consensus has been reached but (a) this would not be guaranteed (frozen non-consensus network states are possible) and (b) add yet further delay in implementing a collective decision, even in cases where the network converged rapidly.

The authors at the end discuss the obvious question of how costly stalemates may be; however the examples they give do not seem to map onto the decision model presented... in the Discussion the authors consider stalemate as 'not going out to forage' for example, but in a binary decision there are only two options-forage or don't forage. So indecision is actually the same as one of the decision options, creating a bias towards that option. This needs considering more thoroughly; the foraging under predation risk example is actually an example of an asymmetric decision problem, in which a quorum decision rule is likely optimal rather than a majority one (see next point). By the way, a decision problem in which indecision is well-defined and can be adaptive is in choices between items, where other options may become available (e.g. Pais et al. <https://doi.org/10.1371/journal.pone.0073216>)

I find the assumption of the majority rule also needs more justification - as the authors are aware, quorum rules can be optimal when decision problems are not purely symmetric (Marshall et al. <https://doi.org/10.7554/eLife.40368>)... actually given the observed similarities between that analysis and the analysis of attention to correlated and uncorrelated cues that the authors extend here (see published review correspondence), I wonder if a non-majority quorum may also qualitatively change the nature of the results from that analysis.

Given the above, while I think this research may interest opinions dynamics researchers, I am not confident that it is informative for biologists.

As a more minor point, the authors use only one random network model; others which are similarly simple to parameterise exist, such as Barabasi-Albert and Erdos-Renyi. I don't think the results will qualitatively change, but it would be good to check their sensitivity to this assumption in online material.

Since I have been quite critical and referred to my own work I sign my review.

Decision letter (RSPB-2020-0096.R0)

31-Jan-2020

Dear Dr Winklmayr:

I am writing to inform you that your manuscript RSPB-2020-0096 entitled "The wisdom of stalemates: consensus and clustering as filtering mechanisms for improving collective accuracy" has, in its current form, been rejected for publication in Proceedings B.

This action has been taken on the advice of referees, who have recommended that substantial revisions are necessary. With this in mind we would be happy to consider a resubmission, provided the comments of the referees are fully addressed. However please note that this is not a provisional acceptance.

Sincerely,
Dr Sasha Dall
mailto: proceedingsb@royalsociety.org

Associate Editor

Comments to Author:

Both reviewers agree that the MS deals with an interesting topic, but both also raise important concerns. Most notably, both think that the cost of a stalemate is not properly taken into account. As addressing an interesting topic, but both raise important concerns. Most notably, both think that the cost of a stalemate is not properly taken into account. As addressing this issue involves modifying the model and redoing some analyses this is not just a cosmetic change.

Reviewer(s)' Comments to Author:

Referee: 1

Comments to the Author(s)

In this study the authors use a modelling approach to explore the potential influence of 'stalemates' on the process of collective decision making. They investigate how consensus decisions based on opinion dynamics perform relative to a simple majority rule in both 'simple' and 'complex' environments and then explore the role of network structure and rewards structure on these patterns.

I found the analysis to be thorough and interesting, and the presentation a rigorous analysis of the concept they present. The idea that stalemates should be considered as an important component of the decision process is certainly worthy of further investigation, and the treatment of network structure effects is revealing. The manuscript is in general well written, though could possibly be improved with some clarification of the modelling approach and results in places.

I do have some concerns with the biological applicability of the model and assumptions.

My greatest concern is with the fundamental premise that stalemates are not costly. I would argue instead that stalemates are, by definition, **always** costly. If an organism is required to make a decision, this implies that the current situation is not suitable and therefore that not deciding will induce a cost. As the decision process (particularly a collective one) is costly in itself, it makes no sense to initiate this process unless there is a cost of not doing so. A recent model of ant decision making during emigrations, for example, assumed that stalemates were equivalent to failure (Cronin 2019 *J. Theor. Biol.*). I agree that the cost of a wrong decision may be higher than the cost of a non-decision, but this is likely why we observe adjustments of quorum thresholds in collective decisions where this the case, to insure against this outcome (eg Dornhaus et al 2004 *Animal Behaviour*).

The authors do make some effort to acknowledge the effect of reward structure in the final part of the manuscript and show that this can strongly influence the patterns observed. I wonder, however, how they can justify their general and strongly stated claims that stalemates improve decision making. In fact, it seems that this is true for only some contexts, and possibly unusual ones at that. I do not think that this detracts from the validity of the results in general, but I do think the authors need to be more cautious and transparent in explaining the context and limitations of their findings.

Minor comments:

I found it unclear of how the correlated information was applied to other individuals in the group. Perhaps this can be made clearer.

How does the 'simple majority' rule work in this simulation? I guess this is simply based on the proportion of initially correct opinions without any iteration?

Opinion dynamics seems to augment the Presumably the opinion polling method would also lead to lower success for lower values of r (<0.5) and that stalemates would then filter out good choices? I think it is worth mentioning this 'dark side' of this method of decision making also relative to the simple majority rule if this is true.

In figure 4, the probability of stalemates appears to have not reached an equilibrium for larger groups, so are we really seeing a 'stalemate' or is this simply an effect of the greater complexity of consensus building in larger groups? This brings into question the definition of a stalemate. Similarly, it is stated that S_{max} is set to "twice the average number of steps for consensus". Was this 'for each group size'? This should be made clear as the average is clearly very different depending on what is averaged.

Referee: 2

Comments to the Author(s)

The authors show that filtering decisions that are likely to be incorrect increases accuracy on the remaining collective decisions; in itself this is obvious, but interestingly the authors use non-convergence of opinion dynamics on networks to achieve the filtering. However I have serious concerns about the biological relevance of the work.

The general question, of how to achieve decisions on networks where there is no centralised tallying of votes is interesting; however, this manuscript begs further questions for me... in

particular, how is consensus or absence thereof to be assessed by group members without precisely such a centralised overseer? I suppose one could assume that group members will monitor their neighbourhood and after a prolonged period of observing no changes of opinion conclude that consensus has been reached but (a) this would not be guaranteed (frozen non-consensus network states are possible) and (b) add yet further delay in implementing a collective decision, even in cases where the network converged rapidly.

The authors at the end discuss the obvious question of how costly stalemates may be; however the examples they give do not seem to map onto the decision model presented... in the Discussion the authors consider stalemate as 'not going out to forage' for example, but in a binary decision there are only two options-forage or don't forage. So indecision is actually the same as one of the decision options, creating a bias towards that option. This needs considering more thoroughly; the foraging under predation risk example is actually an example of an asymmetric decision problem, in which a quorum decision rule is likely optimal rather than a majority one (see next point). By the way, a decision problem in which indecision is well-defined and can be adaptive is in choices between items, where other options may become available (e.g. Pais et al. <https://doi.org/10.1371/journal.pone.0073216>)

I find the assumption of the majority rule also needs more justification - as the authors are aware, quorum rules can be optimal when decision problems are not purely symmetric (Marshall et al. <https://doi.org/10.7554/eLife.40368>)... actually given the observed similarities between that analysis and the analysis of attention to correlated and uncorrelated cues that the authors extend here (see published review correspondence), I wonder if a non-majority quorum may also qualitatively change the nature of the results from that analysis.

Given the above, while I think this research may interest opinions dynamics researchers, I am not confident that it is informative for biologists.

As a more minor point, the authors use only one random network model; others which are similarly simple to parameterise exist, such as Barabasi-Albert and Erdos-Renyi. I don't think the results will qualitatively change, but it would be good to check their sensitivity to this assumption in online material.

Since I have been quite critical and referred to my own work I sign my review.

Review signed by: James Marshall

Author's Response to Decision Letter for (RSPB-2020-0096.R0)

See Appendix A.

RSPB-2020-1802.R0

Review form: Reviewer 2 (James Marshall)

Recommendation

Major revision is needed (please make suggestions in comments)

Scientific importance: Is the manuscript an original and important contribution to its field?
Acceptable

General interest: Is the paper of sufficient general interest?

Good

Quality of the paper: Is the overall quality of the paper suitable?

Acceptable

Is the length of the paper justified?

Yes

Should the paper be seen by a specialist statistical reviewer?

No

Do you have any concerns about statistical analyses in this paper? If so, please specify them explicitly in your report.

No

It is a condition of publication that authors make their supporting data, code and materials available - either as supplementary material or hosted in an external repository. Please rate, if applicable, the supporting data on the following criteria.

Is it accessible?

Yes

Is it clear?

Yes

Is it adequate?

Yes

Do you have any ethical concerns with this paper?

No

Comments to the Author

This paper will probably be well received (i.e. cited).

There are a number of things I find unsatisfying in the revised manuscript, and the authors' replies, and some things I find intriguing.

A large part of my concerns revolve around the Condorcet 'jury theorem' being simultaneously set up as a straw man, and represented as the state of the art in collective decision-making modelling. There are two main problems here: (i) Condorcet is a very simplified view of collective decision-making, with an assumption (of majority decision-making) that is not generally sound. The authors cite Marshall et al. (2019) *eLife* in passing ('by the way, quorums exist') when actually it speaks to a lot of the statements and assumptions of majority decisions made in this manuscript; even if it does not relate directly to the model presented here, repeated championing of Condorcet as the state-of-the-art can only hold the field back (ii) Condorcet is a single shot decision strategy, but here the authors try to move it into the domain of sequential sampling models, which is formally very well developed (dating back to the 1940s). This has already been done in collective decision-making (which the authors do not represent) and even more in individual decision theory, based on fundamental statistical and decision-theoretic principles. It is a disservice to readers, and earlier authors, not to make these connections explicit. In collective decision-making two early efforts (I cannot objectively judge their importance, but they are at least fairly well cited) are Marshall et al. (2009) *J. R. Soc. Interface*, and Pais et al. (2013) *PLoS one* (mentioned in my previous review, but still not cited). These models relate to sequential models, in particular the Drift Diffusion Model (well known in psychology and neuroscience)

and nonlinear attractor models, similarly to how Condorcet relates to signal detection theory as used in psychophysics. These links are important and enriching, in my opinion, because they facilitate reasoning about optimal behaviours. Models such as Pais have led to the value-sensitive decision-making hypothesis, in which adaptive deadlock maintenance or breaking is hypothesised to be adaptive, and mechanisms proposed to do just this, with empirical support to date in different species of primates, and in single-celled organisms; this seems to me to be very relevant to the central thesis of the present manuscript. More recently decision theorists have looked at optimal policies for value-based decision-making, including links to existing decision models such as the DDM (see Tajima et al 2016 and 2019, but also see Marshall (2019) *bioRxiv* and Steverson et al. (2020) *Scientific Reports* for the underlying assumptions, which confirm the optimality of value-sensitive decision-making); it is worth noting that although the present manuscript refers to decision accuracy, it usually describes decision scenarios in which accuracy is not inherently meaningful (actors are rewarded by the value of the option chosen, rather than whether or not they chose the best - see Pirrone et al. 2014 *Frontiers in Neuroscience*). Note that these analyses explicitly cost time in an appropriate manner (e.g. it's not just whether a decision is reached or not, but how long a decision takes to be reached). In short, there's a rich and sophisticated literature in this domain, much of it in collective decision-making, that has been totally ignored.

As an aside, I also found the authors move from 100% consensus to lower quorums to be puzzling - assessing a 100% quorum is probably easier (I suggested a mechanism in my review) than assessing, say, an 80% quorum threshold. I didn't really see this as a resolution of the problem of quorum sensing - it would be better perhaps just to admit this problem and leave its resolution as future work.

The thing I do find interesting is the improvements that can apparently be made to simple wisdom of the crowds (when it is apparently appropriately realised) through opinion dynamics; the advantage in sequential sampling models usually arises because multiple samples of the same random variable decreases uncertainty about its mean; but here it appears there is no repeat sampling. The manuscript attempts this - to the extent that the explanation is not made in terms of degenerate single-shot decision models (I did not filter these I'm afraid) it is potentially useful. Again, however, I suspect this may be usefully related to decision-theoretic analyses from neuroscience, in which opt-out options are available for some decisions and this is usually taken when there is low decision confidence for the agent (see for example, Kianni and Shadlen's papers).

Substantive comments:

line 45: Condorcet makes systematic mispredictions - see Marshall et al. *eLife* (point i above) - see also line 101.

line 53: (point i above) I finally took the trouble to understand how Kao and Couzin's results turned out the way they did, and it is important to understand that they basically represent degenerate parameterisations of collective decision-making models, in which, under the assumption of a majority rule, asymptotic accuracy is not 1, and may even decrease. But there is no reason to suppose groups will use the wrong voting strategy and quorum threshold beyond making arbitrary heuristically-justified strategy set restrictions, and indeed in subsequent work some of the present authors showed how groups could adapt to deploy the optimal voting strategy. Hence I find comparisons of performance improvements against such degenerate decision models to not be informative, unless a very good justification can be given for why the degenerative parameterisations are of interest. Comparing against an (appropriately deployed) wisdom of the crowds model is, however, interesting. Explaining this from first principles, even more so.

Minor comments:

'stalemate filtering' is not in itself very informative, and I'm not sure it's really defined (N.B. *not* 'stalemale' as in the abstract)

Figs 1C and 1D - is 'relative accuracy' correctly placed? Should it not be next to the heat map key? I believe the y-axis is p here (as indicated)

Sorry, this has been possibly a longer review than my original, but the authors response, and time, have helped crystallise for me how what they are doing relates to earlier work. I think collective decision theory is now sufficiently mature that we should be synthesising, rather than coming up with new and apparently surprising results in isolation from each other. My own research project has been to do this in the context of statistical decision theory, hence my focus on that and my own work in my review. I hope the authors will find the criticism constructive, even if not entirely what they may wish to receive.

Decision letter (RSPB-2020-1802.R0)

01-Sep-2020

Dear Ms Winklmayr:

Your manuscript has now been peer reviewed and the reviews have been assessed by an Associate Editor. The reviewers' comments (not including confidential comments to the Editor) and the comments from the Associate Editor are included at the end of this email for your reference. As you will see, the reviewers and the Editors have raised some concerns with your manuscript and we would like to invite you to revise your manuscript to address them.

Research ethics:

Use of animals and field studies:

It is a condition of publication that you make available the data and research materials supporting the results in the article (<https://royalsociety.org/journals/authors/author-guidelines/#data>). Datasets should be deposited in an appropriate publicly available repository and details of the associated accession number, link or DOI to the datasets must be included in the Data Accessibility section of the article (<https://royalsociety.org/journals/ethics-policies/data-sharing-mining/>). Reference(s) to datasets should also be included in the reference list of the article with DOIs (where available).

Please submit a copy of your revised paper within three weeks. If we do not hear from you within this time your manuscript will be rejected. If you are unable to meet this deadline please let us know as soon as possible, as we may be able to grant a short extension.

Best wishes,

Dr Sasha Dall

Associate Editor

Comments to Author:

The reviewer thinks the revision has definitely improved the MS but still has some quite important issues. I think the potential impact would definitely be greater if they are addressed in some form.

Reviewer(s)' Comments to Author:

Referee: 2

Comments to the Author(s).

This paper will probably be well received (i.e. cited).

There are a number of things I find unsatisfying in the revised manuscript, and the authors' replies, and some things I find intriguing.

A large part of my concerns revolve around the Condorcet 'jury theorem' being simultaneously set up as a straw man, and represented as the state of the art in collective decision-making modelling. There are two main problems here: (i) Condorcet is a very simplified view of collective decision-making, with an assumption (of majority decision-making) that is not generally sound. The authors cite Marshall et al. (2019) *eLife* in passing ('by the way, quorums exist') when actually it speaks to a lot of the statements and assumptions of majority decisions made in this manuscript; even if it does not relate directly to the model presented here, repeated championing of Condorcet as the state-of-the-art can only hold the field back (ii) Condorcet is a single shot decision strategy, but here the authors try to move it into the domain of sequential sampling models, which is formally very well developed (dating back to the 1940s). This has already been done in collective decision-making (which the authors do not represent) and even more in individual decision theory, based on fundamental statistical and decision-theoretic principles. It is a disservice to readers, and earlier authors, not to make these connections explicit. In collective decision-making two early efforts (I cannot objectively judge their importance, but they are at least fairly well cited) are Marshall et al. (2009) *J. R. Soc. Interface*, and Pais et al. (2013) *PLoS one* (mentioned in my previous review, but still not cited). These models relate to sequential models, in particular the Drift Diffusion Model (well known in psychology and neuroscience) and nonlinear attractor models, similarly to how Condorcet relates to signal detection theory as used in psychophysics. These links are important and enriching, in my opinion, because they facilitate reasoning about optimal behaviours. Models such as Pais have led to the value-sensitive decision-making hypothesis, in which adaptive deadlock maintenance or breaking is hypothesised to be adaptive, and mechanisms proposed to do just this, with empirical support to date in different species of primates, and in single-celled organisms; this seems to me to be very relevant to the central thesis of the present manuscript. More recently decision theorists have looked at optimal policies for value-based decision-making, including links to existing decision models such as the DDM (see Tajima et al 2016 and 2019, but also see Marshall (2019) *bioRxiv* and Steverson et al. (2020) *Scientific Reports* for the underlying assumptions, which confirm the optimality of value-sensitive decision-making); it is worth noting that although the present manuscript refers to decision accuracy, it usually describes decision scenarios in which accuracy is not inherently meaningful (actors are rewarded by the value of the option chosen, rather than whether or not they chose the best - see Pirrone et al. 2014 *Frontiers in Neuroscience*). Note that these analyses explicitly cost time in an appropriate manner (e.g. it's not just whether a decision is reached or not, but how long a decision takes to be reached). In short, there's a rich and sophisticated literature in this domain, much of it in collective decision-making, that has been totally ignored.

As an aside, I also found the authors move from 100% consensus to lower quorums to be puzzling - assessing a 100% quorum is probably easier (I suggested a mechanism in my review) than assessing, say, an 80% quorum threshold. I didn't really see this as a resolution of the

problem of quorum sensing - it would be better perhaps just to admit this problem and leave its resolution as future work.

The thing I do find interesting is the improvements that can apparently be made to simple wisdom of the crowds (when it is apparently appropriately realised) through opinion dynamics; the advantage in sequential sampling models usually arises because multiple samples of the same random variable decreases uncertainty about its mean; but here it appears there is no repeat sampling. The manuscript attempts this - to the extent that the explanation is not made in terms of degenerate single-shot decision models (I did not filter these I'm afraid) it is potentially useful. Again, however, I suspect this may be usefully related to decision-theoretic analyses from neuroscience, in which opt-out options are available for some decisions and this is usually taken when there is low decision confidence for the agent (see for example, Kianni and Shadlen's papers).

Substantive comments:

line 45: Condorcet makes systematic mispredictions - see Marshall et al. eLife (point i above) - see also line 101.

line 53: (point i above) I finally took the trouble to understand how Kao and Couzin's results turned out the way they did, and it is important to understand that they basically represent degenerate parameterisations of collective decision-making models, in which, under the assumption of a majority rule, asymptotic accuracy is not 1, and may even decrease. But there is no reason to suppose groups will use the wrong voting strategy and quorum threshold beyond making arbitrary heuristically-justified strategy set restrictions, and indeed in subsequent work some of the present authors showed how groups could adapt to deploy the optimal voting strategy. Hence I find comparisons of performance improvements against such degenerate decision models to not be informative, unless a very good justification can be given for why the degenerative parameterisations are of interest. Comparing against an (appropriately deployed) wisdom of the crowds model is, however, interesting. Explaining this from first principles, even more so.

Minor comments:

'stalemate filtering' is not in itself very informative, and I'm not sure it's really defined (N.B. *not* 'stalemate' as in the abstract)

Figs 1C and 1D - is 'relative accuracy' correctly placed? Should it not be next to the heat map key? I believe the y-axis is p here (as indicated)

Sorry, this has been possibly a longer review than my original, but the authors response, and time, have helped crystallise for me how what they are doing relates to earlier work. I think collective decision theory is now sufficiently mature that we should be synthesising, rather than coming up with new and apparently surprising results in isolation from each other. My own research project has been to do this in the context of statistical decision theory, hence my focus on that and my own work in my review. I hope the authors will find the criticism constructive, even if not entirely what they may wish to receive.

Author's Response to Decision Letter for (RSPB-2020-1802.R0)

See Appendix B.

Decision letter (RSPB-2020-1802.R1)

02-Oct-2020

Dear Ms Winklmayr

I am pleased to inform you that your manuscript RSPB-2020-1802.R1 entitled "The wisdom of stalemates: consensus and clustering as filtering mechanisms for improving collective accuracy" has been accepted for publication in Proceedings B.

The referee(s) have recommended publication, but also suggest some minor revisions to your manuscript. Therefore, I invite you to respond to the referee(s)' comments and revise your manuscript. Because the schedule for publication is very tight, it is a condition of publication that you submit the revised version of your manuscript within 7 days. If you do not think you will be able to meet this date please let us know.

Sincerely,
Dr Sasha Dall
Editor, Proceedings B
<mailto:proceedingsb@royalsociety.org>

Associate Editor:
Board Member
Comments to Author:

I think most criticisms have been adequately dealt with, and I don't see much benefit resulting from more back-and-forths with the reviewers. I do like the MS myself, but there are still some issues that might clarify some things. First of all, when rereading the MS, I kept wondering about the relative cost of a stalemate situation. This is discussed later on, but a few words in the beginning, where it is mentioned that a stalemate is effectively a third option would have helped me. I think it also should be more clear what 'accuracy' means in a three-option outcome. And finally I do not understand the sentence on ll. 100-101...

Author's Response to Decision Letter for (RSPB-2020-1802.R1)

See Appendix C.

Decision letter (RSPB-2020-1802.R2)

09-Oct-2020

Dear Ms Winklmayr

I am pleased to inform you that your manuscript entitled "The wisdom of stalemates: consensus and clustering as filtering mechanisms for improving collective accuracy" has been accepted for publication in Proceedings B.

Open Access

Paper charges

Sincerely,

Dr Sasha Dall

Associate Editor:

Comments to Author:

Very good! I see no further issues.

Appendix A

“The wisdom of stalemates: consensus and clustering as filtering mechanisms for improving collective accuracy” (ref: RSPB-2020-0096)

Detailed response to the Editor and the Referees

We wish to thank the referees for their thorough and constructive comments which provided helpful insights and allowed us to further clarify and deepen our original results.

The present version of the manuscript contains substantial changes and additional analysis. In particular, we shortened the first two parts of the results section and unified Figures 1 and 2. We moved the section about limited decision speed to the supplementary material where we also discuss the applicability of our findings to other theoretical and real-world networks. In order to address the concerns raised by the referees we added a new section (“Detecting and Breaking Stalemates”) complete with a new Figure (Figure 3 in the present manuscript), where we discuss how groups can assess that a stalemate has occurred and how they can escape it. Finally, we also revised the section about reward structure and the respective Figure (previously Figure 5, now Figure 4) to more clearly support our claim that stalemates can be beneficial even when they impose a cost. The present version of the manuscript also includes five short supplementary sections where we extend our results to cover not only other theoretical graphs, but also biological networks from experimental data and compare our results about stalemate filtering to other known voting strategies.

In the following we list the referees’ comments (green for Referee 1 and blue for Referee 2) together with our responses (in black).

Referee 1:

(...) The manuscript is in general well written, though could possibly be improved with some clarification of the modeling approach and results in places.

I do have some concerns with the biological applicability of the model and assumptions.

My greatest concern is with the fundamental premise that stalemates are not costly. I would argue instead that stalemates are, by definition, *always* costly. If an organism is required to make a decision, this implies that the current situation is not suitable and therefore that not deciding will induce a cost. As the decision process (particularly a collective one) is costly in itself, it makes no sense to initiate this process unless there is a cost of not doing so. A recent model of ant decision making during emigrations, for example, assumed that stalemates were equivalent to failure (Cronin 2019 J. Theor. Biol). I agree that the cost of a wrong decision may be higher than the cost of a non-decision, but this is likely why we observe adjustments of quorum thresholds in collective decisions where this the case, to insure against this outcome (eg Dornhaus et al 2004 Animal Behaviour).

The authors do make some effort to acknowledge the effect of reward structure in the final part of the manuscript and show that this can strongly influence the patterns observed. I wonder, however, how they can justify their general and strongly stated claims that stalemates improve decision making. In fact, it seems that this is true for only some contexts, and possibly unusual ones at that. I do not think that this detracts from the validity of the results in general, but I do think the authors need to be more cautious and transparent in explaining the context and limitations of their findings.

We agree with Referee 1 that stalemates are always costly; at the very least, they incur some nonzero time, and therefore energetic, cost. Indeed, for some contexts and species, stalemates can be quite costly. For example, in (Cronin, 2019) that Referee 1 referenced, ant colonies were forced to emigrate to a new nest site; in this context, the cost of stalemate is high -- the queen and brood are highly vulnerable and failure to emigrate quickly can lead to desiccation of brood or attack by others.

However, we do think that it is not unusual for the cost of stalemates to be negligible compared to the cost of a wrong decision. We described in detail one such example (lines 34-40): small schooling fish which spend most of their time hiding in shelters and minimize their time spent in open space to decrease their risk of predation. The cost of being eaten, in this case, is much greater than the time/energetic cost of continuing to wait in the shelter. We think that this context is common for many small prey species. Another example is the initiation of a migration by an animal group. Migrating too early in the season, or in the wrong direction, can be highly costly to individuals in the group, leading to higher mortality during the migration, or failure to reach their desired destination. Here again, the time/energetic cost of delaying the onset of migration can be negligible compared to the cost of a wrong decision.

Since there are many contexts in which the cost of a stalemate is small compared to the cost of a wrong decision, we think that it is justified to first consider the special case where there is zero cost for a stalemate. Not only is this a reasonable assumption for many natural cases, it is also a simpler scenario to analyze for the purposes of explaining to the reader the core mechanism and phenomenon underlying stalemates during collective decision-making.

We have clarified why we think that stalemates have negligible costs in many contexts and when they incur greater costs (lines 34-40, 207-213). Also, in the new section "Detecting and Breaking Stalemates" we explicitly acknowledge that a stalemate will ultimately have to be overcome by either changing the decision strategy or the use of additional information. From this point of view a stalemate can be seen as a delay in the decision process which accumulates a temporal cost until a consensus is reached. In the revised section "The reward structure modulates the optimal collective decision-making process" and Figure 4 of the present manuscript we explicitly examine the cost of stalemates as a time cost, and the results suggests that stalemates can be beneficial to decision making even when they incur a non-zero cost.

Minor comments:

I found it unclear of how the correlated information was applied to other individuals in the group. Perhaps this can be made clearer.

We have revised our description of the model to make this clearer (lines 102-107).

How does the 'simple majority' rule work in this simulation? I guess this is simply based on the proportion of initially correct opinions without any iteration?

This is correct -- we have revised our main text to make this clearer (lines 109-110).

Opinion dynamics seems to augment the Presumably the opinion polling method would also lead to lower success for lower values of r (<0.5) and that stalemates would then filter out good choices? I think it is worth mentioning this 'dark side' of this method of decision making also relative to the simple majority rule if this is true.

Yes, this is theoretically true. However, because we are specifically focused on biological organisms, we think it is reasonable to assume that the cues that the individuals attune to tend to be beneficial to them ($r > 0.5$). If the individuals happen to attune to a cue for which $r < 0.5$, then learning, or evolution, would be likely to quickly reverse individuals' interpretation of the cue, such that it becomes a positively informative cue. Nonetheless, we've briefly mentioned the theoretical possibility of this 'dark side' of stalemates (lines 130-131).

In figure 4, the probability of stalemates appears to have not reached an equilibrium for larger groups, so are we really seeing a 'stalemate' or is this simply an effect of the greater complexity of consensus building in larger groups? This brings into question the definition of a stalemate. Similarly, it is stated that S_{max} is set to "twice the average number of steps for consensus". Was this 'for each group size'? This should be made clear as the average is clearly very different depending on what is averaged.

We have updated Figure 4 to include longer sampling times, the new version can be found in section S2 of the supplementary material. We also made more explicit in which way we use S_{max} to detect stalemates in simulations (lines 90-94). Additionally, we provide an overview of saturation times for different networks in section S1 of the SI.

Referee: 2

Comments to the Author(s)

The authors show that filtering decisions that are likely to be incorrect increases accuracy on the remaining collective decisions; in itself this is obvious, but interestingly the authors use non-convergence of opinion dynamics on networks to achieve the filtering. However I have serious concerns about the biological relevance of the work.

The general question, of how to achieve decisions on networks where there is no centralised tallying of votes is interesting; however, this manuscript begs further questions for me... in particular, how is consensus or absence thereof to be assessed by group members without precisely such a centralised overseer? I suppose one could assume that group members will monitor their neighbourhood and after a prolonged period of observing no changes of opinion conclude that consensus has been reached but (a) this would not be guaranteed (frozen non-consensus network states are possible) and (b) add yet further delay in implementing a collective decision, even in cases where the network converged rapidly.

This is a valid criticism. For the sake of expositional clarity, we think that it's still useful to first consider the case of having a threshold of full consensus (i.e., 100% of the

opinions need to agree for a group for a consensus to have been reached). This case is the simplest, and most clearly reveals the fundamental mechanisms at play that allows stalemates to improve collective wisdom.

However, we agree with Referee 2 that it may be difficult for real animals in groups to adjudge whether a full consensus has been reached -- and this ability surely varies from species to species. For example, the mechanism allowing animals in groups to effectively 'vote' can take many forms -- some examples are listed in Table 1 in reference 12 in our manuscript (Conradt & Roper, 2003). In addition, the Supporting Information of one of the present author's papers (Kao, et al., 2014); reference 19 in our manuscript) demonstrates that even standard schooling models can reliably produce consensus.

In the examples cited above, a group 'decision' tends to occur when a majority for one option has been reached (simple majority rule). However, there are other examples from nature of a group decision occurring when a threshold other than 50% has been reached (a quorum threshold; nicely summarized in the discussion of (Marshall, et al., 2019)). The criticism from Referee 2 therefore appears to be that a quorum threshold of 100% (i.e., full consensus) may be difficult for real animals to implement, compared to a less extreme threshold. Referee 2 hypothesized one possible way in which this may be achieved locally (at least approximately). Rather than speculating on possible ways to locally detect full consensus (which may well vary across species), we instead adopted a different strategy to address this criticism. We re-ran our model but required a threshold less than 100% to qualify as 'consensus.' As shown in our revised results (new section "Detecting and Breaking Stalemates" as well as the new Figure 3), our general result that stalemates can improve collective wisdom still holds but the effect is weaker (unsurprisingly). Therefore, our new analysis shows that animal groups can still benefit from the mechanism of stalemate filtering even if they imperfectly detect consensus.

We note that the 'quorum threshold' that we use to judge consensus is similar to, but technically not the same as, the quorum threshold employed by Marshall et al and related papers. In the latter case, the group is predisposed towards one of the two options and only selects the other option if the quorum threshold has been reached. In our case, the group is predisposed towards a stalemate unless the quorum threshold has been reached for either of the two options. Nonetheless, in both cases, the group needs some mechanism to determine whether the threshold has been reached. In our new analysis, we showed that this threshold need not be 100% for groups to enjoy some of the benefits of stalemate filtering.

The authors at the end discuss the obvious question of how costly stalemates may be; however the examples they give do not seem to map onto the decision model presented... in the Discussion the authors consider stalemate as 'not going out to forage' for example, but in a binary decision there are only two options-forage or don't forage. So indecision is actually the same as one of the decision options, creating a bias towards that option. This needs considering more thoroughly; the foraging under predation risk example is actually an example of an asymmetric decision problem, in which a quorum decision rule is likely optimal rather than a majority one (see next point). By the way, a decision problem in which indecision is well-defined and can be adaptive is in choices between items, where other options may become available (e.g. Pais et al. <https://doi.org/10.1371/journal.pone.0073216>)

I find the assumption of the majority rule also needs more justification - as the authors are aware, quorum rules can be optimal when decision problems are not purely symmetric (Marshall et al. <https://doi.org/10.7554/eLife.40368>)... actually given the observed similarities between that analysis and the analysis of attention to correlated and uncorrelated cues that the authors extend here (see published review correspondence), I wonder if a non-majority quorum may also qualitatively change the nature of the results from that analysis.

The decision scenario that we study here is not exactly the same as the asymmetric decision problem that Referee 2 mentioned. In fact, here we have three potential options that the group can select from: indecision (i.e., stalemate), a consensus decision for option A (e.g., one potential food patch), and a consensus decision for option B (e.g., a second potential food patch) -- this is described in the first paragraph of the Results. Each of these three possibilities has a reward associated with it (which we explicitly examined in the 'reward structure' subsection of the Results, and in earlier subsections simply assume that the cost of stalemates are negligible compared to the costs and rewards of an incorrect and correct consensus; see response to Referee 1 for a justification of this).

There does exist a general solution to the problem of polychotomous choice scenarios (where there are >2 options available), found in (Ben-Yashar & Paroush, 2001). However, our scenario is also different from the one described in that paper because, in our model, while the group collectively can select from three possible options, each individual in the group is only aware of two options (they can form an opinion for either option A or B, never for a stalemate). Therefore, stalemates can only emerge at the collective level and are not represented at the individual level.

Because of this, it appears that the dichotomous choice solution is still relevant for our particular case (between option A and option B), and we can calculate the optimal quorum threshold for any particular environment. However, since this calculation does not take into account stalemates at all, we applied this comparison only to the case where stalemates are costless. In our new analysis (section S5 in the SI), we compared three collective decision-making strategies: 1) a one-shot decision process in a complex environment where a collective decision is made by simple majority 2) a one-shot simple majority decision process where individuals use both the uncorrelated and the correlated cues, following the optimal individual-level voting behavior (described in the Results (section b) of (Kao, et al., 2014) this can be thought of as an individual level quorum threshold), and 3) individuals follow the optimal individual-level voting behavior, but the individuals use opinion dynamics to form a consensus (or stalemate). This comparison reveals more clearly that stalemates can improve decision accuracy, even compared to the optimal quorum threshold.

Indeed, the mechanism of stalemate filtering is orthogonal to that of other collective strategies, including a quorum threshold, and can be seen as an additional tool that group-living animals may utilize, on top of other tools, to improve the quality of their decisions. In other words, quorum thresholds and stalemates are not mutually exclusive strategies, but are simultaneously available to be used by animal groups. Since the beneficial role of stalemates has thus far not been highlighted (especially compared to the work done on quorum thresholds), we think that this paper will be

informative for biologists by highlighting a mechanism that animal groups may well be exploiting but hasn't been recognized by researchers.

Given the above, while I think this research may interest opinions dynamics researchers, I am not confident that it is informative for biologists.

As a more minor point, the authors use only one random network model; others which are similarly simple to parameterise exist, such as Barabasi-Albert and Erdos-Renyi. I don't think the results will qualitatively change, but it would be good to check their sensitivity to this assumption in online material.

This is a valid point. We chose the Watts-Strogatz family of networks because it best captures some of the essential features of animal group networks (many local connections with some long range connections), but for the sake of completeness (and sensitivity) it is useful to examine other network families as well. We omit simulating the Erdos-Renyi network, since the Watts-Strogatz family of networks includes random networks if $\beta = 1$ (see Figure 2A). In a new set of analyses, we reproduced our analyses for Barabasi-Albert networks, and found that in such networks groups are almost certain to reach consensus which diminishes the benefits of stalemate filtering, except for extremely sparsely connected graphs (see section S3 and Figure S2 in the supplementary material). Additionally, we also studied the effect of stalemate filtering in realistic biological networks, inferred from position data of schooling fish where we could again confirm the beneficial effect of stalemate filtering on collective accuracy (see section S4 and Figure S3 in the supplementary material).

Again, we want to thank the referees for their very constructive comments that helped to expand and improve our original results. We hope that the manuscript in its current form answers the valid concerns raised by the referees and can be reconsidered for publication.

Yours sincerely,

Claudia Winklmayr, Albert B. Kao, Joseph Bak-Coleman and Pawel Romanczuk

Appendix B

“The wisdom of stalemates: consensus and clustering as filtering mechanisms for improving collective accuracy”

ref: RSPB-2020-0096

Detailed response to the Editor and the Referees

We wish to thank the referee for his detailed comments, especially for pointing out the interesting connection to value-based decision making. In response to the referee's comments we have made a number of changes to the manuscript. We have moved the part of the results section, where we discussed reward (“The reward structure modulates the optimal collective decision-making process”) and the corresponding Figure (previously Figure 4) to the SI. Thereby we have gained space to include a short discussion of the literature on value-based decision making and how it could be related to our model. To address the reviewer's concerns regarding our treatment of the Condorcet jury theorem, we have added a paragraph to the introduction where we review the limitations of this classic result. In the following we list the referee's comments in color together with our responses in black.

Comments to the Author(s).

This paper will probably be well received (i.e. cited).

There are a number of things I find unsatisfying in the revised manuscript, and the authors' replies, and some things I find intriguing.

A large part of my concerns revolve around the Condorcet 'jury theorem' being simultaneously set up as a straw man, and represented as the state of the art in collective decision-making modelling. There are two main problems here: (i) Condorcet is a very simplified view of collective decision-making, with an assumption (of majority decision-making) that is not generally sound. The authors cite Marshall et al. (2019) eLife in passing ('by the way, quorums exist') when actually it speaks to a lot of the statements and assumptions of majority decisions made in this manuscript; even if it does not relate directly to the model presented here, repeated championing of Condorcet as the state-of-the-art can only hold the field back

We acknowledge the detailed criticism by the referee, and in particular his worry that by referring to the Condorcet jury theorem, we give it too much credit and generate the impression that it is state-of-the-art in collective-decision making. This is certainly not our intention, and we agree with the reviewer that the Condorcet jury theorem, being over 200 years old, is outdated and that much work in the past 30 years has been done to examine relaxations of this classic model's assumptions. In our previous revision, we cited a sampling of such papers, such as Boland “Majority systems and the Condorcet jury theorem” (1989), which relaxes both the independence assumption and the assumption that individuals have equal competence; and Ben-Yashar and Nitzan “The optimal decision rule for fixed-size committees in dichotomous choice situations” (1997), which examines, among other things, how quorum thresholds are optimal depending on the relative cost/rewards of the different options. In our current version, we have rephrased parts of our Introduction (lines 23-24, 33-36) make this point clearer and especially highlighted papers that have relaxed the assumptions of the Condorcet theorem, including the papers above, as well as the reviewer's own work that he mentioned.

Nonetheless, we think that to gain useful insight into the interaction between opinion dynamics and collective wisdom, it is wise to begin with simple decision scenarios. Therefore, while the Condorcet jury theorem is not state-of-the-art in its own right, it is a useful starting point for our work. However, in our work we also examine not only the Condorcet scenario, but a slightly more complex scenario involving a mixture of correlated and independent information, which relaxes the assumption of independence (and can also be interpreted as setting different prior probabilities of the decision outcomes). We now describe our rationale for examining these scenarios in lines 66-71.

Furthermore, we don't consider the use of the Condorcet theorem, or more precisely, majority rule, to be an unfair straw man for comparison. While many animal groups, especially the social insects that the reviewer has studied, do employ quorum rules to make collective decision, majority decisions are still thought to be a relatively accurate description of decision making in many other social animal species (e.g., Strandburg-Peshkin et al "Shared decision-making drives collective movement in wild baboons" (2015) and Couzin et al "Uninformed individuals promote democratic consensus in animal groups" (2011)). Therefore, we feel that comparing the outcome of our opinion dynamics models to majority rule is a fair, and biologically motivated, comparison to make.

In summary, we hope that these revisions to our Introduction help emphasize that the Condorcet jury theorem is not state-of-the-art, highlight work that goes beyond Condorcet, and justify why we still think that it is a useful starting point for our particular work on opinion dynamics.

(ii) Condorcet is a single shot decision strategy, but here the authors try to move it into the domain of sequential sampling models, which is formally very well developed (dating back to the 1940s). This has already been done in collective decision-making (which the authors do not represent) and even more in individual decision theory, based on fundamental statistical and decision-theoretic principles. It is a disservice to readers, and earlier authors, not to make these connections explicit.

The majority of our present work remains within the domain of single shot decision strategies, with sequential sampling only briefly touched upon. Indeed, our main focus is the role of stalemates emerging from local interactions on a network. In the simplest scenario, we consider agents that sample the environment only once - in analogy to single shot decision processes - and then, instead of a global or mean-field aggregation procedure (simple majority or quorum), interact locally on a network through opinion dynamics interactions to try to come to a consensus. In the opinion dynamics interactions, there is no sequential sampling of external information performed by the agents. Each agent switches its opinion (e.g. individual decision) directly based on its network neighborhood.

Later (in the section "Detecting and Breaking Stalemates"), we discuss redrawing of environmental cues, which could be potentially linked to sequential models, when we investigate how stalemates could be resolved. But even here the connection is not straightforward, as implied by the referee. We would like to emphasize that due to the microscopic agent-based formulation of our model on a network, it differs fundamentally from low-dimensional population models of neural (Bogacz et al, 2007,

Trends in Cog Sci) or collective decision making (Pais et al 2013, see also below), where such mapping could be performed.

Because the domain of sequential sampling models relates only to one part of our results (and a minor part at that), and because our manuscript is already at the journal page limits, we think that a discussion of sequential sampling models is beyond the scope of our present work. However, we now briefly highlight the existence of other models and mechanisms of collective decision-making in our Introduction (lines 23-24, 33-36).

In collective decision-making two early efforts (I cannot objectively judge their importance, but they are at least fairly well cited) are Marshall et al. (2009) *J. R. Soc. Interface*, and Pais et al. (2013) *PLoS one* (mentioned in my previous review, but still not cited). These models relate to sequential models, in particular the Drift Diffusion Model (well known in psychology and neuroscience) and nonlinear attractor models, similarly to how Condorcet relates to signal detection theory as used in psychophysics. These links are important and enriching, in my opinion, because they facilitate reasoning about optimal behaviours.

While a general discussion of sequential sampling models is beyond the scope of our paper, we acknowledge that the paper by Pais et al., which the referee is a co-author on, may have some interesting conceptual links to our work.

We first note some fundamental differences of our model to the one studied in Pais et al. 2013:

- 1) In Pais et al 2013, a coarse-grained, infinite population model with noise is considered. In our work we consider an agent-based (“microscopic”) model on a network. Coarse-graining microscopic models is extremely challenging, and rarely possible without drastic simplification. Here, it is not even clear how/whether our model would map to something similar in structure to the equations of Pais et al. on the macroscopic level. Of course, in the end all dynamical (collective) decision making models with two options must eventually, in the lowest order, map to a normal form equations of a pitchfork bifurcation. This is a must, given the symmetries and constraints of the problem, even if the mapping from complex agent-based models may be not feasible analytically. For example, Daniels and Romanczuk, discuss this for a simple neuronal integrator model (Daniels & Romanczuk, 2020, <https://arxiv.org/pdf/1903.09710>). However, the normal form is in itself only of very limited use without knowing the explicit mapping, as one simply does not know how the various microscopic parameters enter the normal form parameters. In a complex agent-based system, single microscopic parameters may at the same time affect the deterministic as well as the stochastic part of coarse-grained equation. An example of this is provided already by the analysis of a simple flocking model in 1d (Romanczuk and Erdmann, 2010, <https://link.springer.com/content/pdf/10.1140/epjst/e2010-01277-0.pdf>), which is mathematically closely related to binary decision making. There the microscopic social interaction strength μ affects both the resulting average speed (deterministic part) and the temperature of the system (effective noise strength in finite systems).

2) Pais et al. 2013 consider explicitly continuous resampling of the options with a finite rate until a decision is made. This is in stark contrast to our model, in which individuals typically sample from the environment only once (see above).

3) The model in Pais et al. 2013 includes a specific cross-inhibiting social interaction mechanism (“stop signaling”), identified in a seminal work in microscopic interactions underlying collective-decision making in bees. This is different from the opinion dynamics mechanism in our model.

4) Pais et al. 2013 consider options with continuous values v , which is also not the case in our model. In our model we have a binary decision between option A and B with unreliable environmental cues. In principle, one could argue that reliability of cues may be interpreted to “quality” or “value” in an abstract sense, however we feel discussing such a superficial link would rather confuse readers without contributing to understanding of core results we want to focus on. It is certainly interesting but should in our opinion be done properly with solid mathematical theoretical underpinning elsewhere.

Despite the fundamental differences between our work and Pais et al., there is the interesting possibility of a deadlock/indecision state in Pais et al. for two low value options, which appears conceptually related to the stalemate filtering in our model. In order to acknowledge this, we have extended our discussion section by discussing this connection (see lines 229-239).

Models such as Pais have led to the value-sensitive decision-making hypothesis, in which adaptive deadlock maintenance or breaking is hypothesised to be adaptive, and mechanisms proposed to do just this, with empirical support to date in different species of primates, and in single-celled organisms; this seems to me to be very relevant to the central thesis of the present manuscript. More recently decision theorists have looked at optimal policies for value-based decision-making, including links to existing decision models such as the DDM (see Tajima et al 2016 and 2019, but also see Marshall (2019) biorXiv and Steverson et al. (2020) Scientific Reports for the underlying assumptions, which confirm the optimality of value-sensitive decision-making); it is worth noting that although the present manuscript refers to decision accuracy, it usually describes decision scenarios in which accuracy is not inherently meaningful (actors are rewarded by the value of the option chosen, rather than whether or not they chose the best - see Pirrone et al. 2014 Frontiers in Neuroscience). Note that these analyses explicitly cost time in an appropriate manner (e.g. it's not just whether a decision is reached or not, but how long a decision takes to be reached). In short, there's a rich and sophisticated literature in this domain, much of it in collective decision-making, that has been totally ignored.

We thank the referee for pointing out the important contributions made in the field of value-based decision making. In order to respect the rich and sophisticated literature on value-sensitive decision-making, we have moved our previously final section on the effect of opinion dynamics in decisions where the costs/rewards of different decision options are varied, to the SI, since we were not able to do a comprehensive investigation of this extension in the present manuscript due to space constraints. This then bought us space in our manuscript to draw connections between our present work and the work described by the reviewer (including several of his own papers that he mentions) on value-based decision making –(see lines 229-239, 249-262). However,

we also want to point out that while this work certainly provides many relevant and important insights, the basic approach differs quite strongly from the model we set out to study. Most of the models employed in value-based decision problems are drift diffusion models where the role of other group members in affecting a focal individual's choice, is captured by a mean-field approach. In our work, however, we explicitly consider the effect of network structure and opinion distribution in small to intermediate sized networks.

Overall, we acknowledge the referee's broad expert knowledge on the theory of (collective) decision making, and his obvious commitment to advance the field further. Given the strict length limits of this journal, we are unable to comprehensively draw links from our paper to all other relevant subfields. With the space that we have available, we have now drawn stronger links, especially to the topics and work that the reviewer has mentioned. In general, we would be very much excited about a general discussion on the theory of (collective) decision making being laid out in a more appropriate setting, such as a review or opinion article, for example by the referee or a group of authors bringing together different perspectives/approaches.

We hope that this dual move -- moving one section of our results to the SI and using the gained space to more generally discuss value-sensitive decision-making -- is a fair way to acknowledge the existence and importance of this area of research, and point readers in this direction as a topic of future work.

As an aside, I also found the authors move from 100% consensus to lower quorums to be puzzling - assessing a 100% quorum is probably easier (I suggested a mechanism in my review) than assessing, say, an 80% quorum threshold. I didn't really see this as a resolution of the problem of quorum sensing - it would be better perhaps just to admit this problem and leave its resolution as future work.

In the referee's previous review, he did mention a possible mechanism, writing: "I suppose one could assume that group members will monitor their neighbourhood and after a prolonged period of observing no changes of opinion conclude that consensus has been reached but (a) this would not be guaranteed (frozen non-consensus network states are possible) and (b) add yet further delay in implementing a collective decision, even in cases where the network converged rapidly."

As the reviewer mentioned, this hypothetical mechanism isn't failproof, and individuals could believe that there is a global consensus when, in fact, there is not one. Overall, we think that the relevant point here is not what the exact mechanism group members could use to assess consensus (as we pointed out in our previous reply, this likely depends on the particular social species under study), but the fact that any local estimation of consensus without a centralized overseer will likely only imperfectly measure global consensus.

The examples in Figure 3A and 3B thus do not represent examples of a group correctly identifying an 80% or 90% quorum (which we agree would be much harder to do) but rather examples of a group behaving as *if* a full consensus had been reached, when in fact it has not. Our results then show that even in the case of such a misjudgment, the beneficial effects of stalemate filtering, although quantitatively reduced, do still apply qualitatively. We changed lines 189-192 for clarifications.

Our analysis of the effect of imperfect estimations of global consensus by examining quorum thresholds below 100% (which would be the case, for example, when there is a frozen non-consensus network state, as the reviewer mentioned) is, we feel, a more general treatment of the robustness of our results. In short, we refrain from specifying *how* consensus is estimated (since this may be species-dependent) and examine the broader question of what happens when consensus is only imperfectly estimated.

The thing I do find interesting is the improvements that can apparently be made to simple wisdom of the crowds (when it is apparently appropriately realised) through opinion dynamics; the advantage in sequential sampling models usually arises because multiple samples of the same random variable decreases uncertainty about its mean; but here it appears there is no repeat sampling. The manuscript attempts this - to the extent that the explanation is not made in terms of degenerate single-shot decision models (I did not filter these I'm afraid) it is potentially useful. Again, however, I suspect this may be usefully related to decision-theoretic analyses from neuroscience, in which opt-out options are available for some decisions and this is usually taken when there is low decision confidence for the agent (see for example, Kianni and Shadlen's papers).

We are very happy that the referee finds the main result of our paper interesting. However, we would disagree that we do explain anything in terms of “degenerate single-shot decisions”, but rather used the widely known simple majority voting as a “null model” for comparison. Regarding the connection to the decision-theoretic analyses in neuroscience, we are more than happy to establish a corresponding connection, which we attempt by a revision of the introduction at lines 33-26 and the discussion at lines 257-262. However, we note that the work by Kanni and Shadlen, *Science* (2009) presents only experimental results on recordings of the firing rate of LIF neurons, and discusses them in terms of a Bayesian framework consistent with a single agent drift-diffusion decision model. There is no discussion on proximate mechanism for distributed collective decision making, which is the focus of our work. Here, from a computational neuroscience perspective other papers are actually more relevant, as e.g. Daniels, Flack, Krakauer, *Front. Neurosci* (2017).

Substantive comments:

line 45: Condorcet makes systematic mispredictions - see Marshall et al. *eLife* (point i above) - see also line 101.

Please see our treatment of the Condorcet jury theorem and related work above.

line 53: (point i above) I finally took the trouble to understand how Kao and Couzin's results turned out the way they did, and it is important to understand that they basically represent degenerate parameterisations of collective decision-making models, in which, under the assumption of a majority rule, asymptotic accuracy is not 1, and may even decrease. But there is no reason to suppose groups will use the wrong voting strategy and quorum threshold beyond making arbitrary heuristically-justified strategy set restrictions, and indeed in subsequent work some of the present authors showed how groups could adapt to deploy the optimal voting strategy. Hence I find comparisons of performance improvements against such degenerate decision models to not be informative, unless a very good justification can be given for why the

degenerative parameterisations are of interest. Comparing against an (appropriately deployed) wisdom of the crowds model is, however, interesting. Explaining this from first principles, even more so.

As we wrote in our previous reply, our mechanism of statement filtering is orthogonal to that of other collective strategies, and can be seen as an additional tool that group-living animals may utilize, on top of other tools, to improve the quality of their decisions. Stalemates appear to improve decision accuracy independent of other parameters in the model. In the parameter sweep (Figure 1C and 1D) that we performed for our previous revision, we examined decision making through opinion dynamics, and decision making through majority rule, using the same underlying voting strategy (whether optimal or not). Therefore, we argue that our analysis is more comprehensive than that suggested by the reviewer, which would have restricted our analysis to only the subset of parameters that are optimal. We found that stalemates improve accuracy under optimal conditions, and also under suboptimal conditions.

Last but not least, we believe that our focus on local (proximate) mechanisms for improvement of collective decision making, independent whether it is optimal or sub-optimal, is also advisable from an evolutionary ecology point of view. Focusing solemnly on optimality of collective decision making at the macro level, implicitly assumes group-level selection. This may be reasonable for collectives of closely related individuals as e.g. colonies of eusocial insects, but in general does not apply to fission-fusion groups consisting of non-related individuals, such as bird flocks or fish schools. It is rather the rule than the exception that evolutionary stable strategies are sub-optimal with respect to the macro-level behavior, and corresponding social dilemmas are being widely studied in evolutionary game theory. In a recent preprint one of the authors discussed explicitly such a social dilemma in the context of collective predator evasion (Klamser and Romanczuk, <https://arxiv.org/abs/2009.02079>).

Minor comments:

'stalemate filtering' is not in itself very informative, and I'm not sure it's really defined (N.B. *not* 'stalemale' as in the abstract)

For more clarity we formally introduce the term 'stalemate filtering' in the abstract as well as in the results section (line 135).

Figs 1C and 1D - is 'relative accuracy' correctly placed? Should it not be next to the heat map key? I believe the y-axis is p here (as indicated)

We have removed the 'relative accuracy' label from Figures 1C and 1D to avoid confusion.

Sorry, this has been possibly a longer review than my original, but the authors response, and time, have helped crystallise for me how what they are doing relates to earlier work. I think collective decision theory is now sufficiently mature that we should be synthesising, rather than coming up with new and apparently surprising results in isolation from each other. My own research project has been to do this in the context of statistical decision theory, hence my focus on that and my own work in my review. I hope the authors will find the criticism constructive, even if not entirely what they may wish to receive.

Again, we want to thank the referee for his extensive and thorough review of our work. We hope that the manuscript in its current form answers the concerns raised by the referee and can be reconsidered for publication.

Yours sincerely,

Claudia Winklmayr, Albert B. Kao, Joseph Bak-Coleman and Pawel Romanczuk

Appendix C

“The wisdom of stalemates: consensus and clustering as filtering mechanisms for improving collective accuracy”

ref: RSPB-2020-1802.R1

Detailed response to the Editor and the Referees

We wish to thank the editor and the referees for accepting the manuscript and identifying some unclear points. In the present version of the manuscript we aim to address these issues, which we hope makes our arguments more precise. In the following we list the referee's comments in color together with our responses in black:

I think most criticisms have been adequately dealt with, and I don't see much benefit resulting from more back-and-forths with the reviewers. I do like the MS myself, but there are still some issues that might clarify some things. First of all, when rereading the MS, I kept wondering about the relative cost of a stalemate situation. This is discussed later on, but a few words in the beginning, where it is mentioned that a stalemate is effectively a third option would have helped me.

We have changed lines 38-40 and lines 45-47 of the introduction to make clear that the cost of a stalemate is an important aspect of the decision problem. We also make more transparent why we decide not to focus on stalemate costs for the first part of the manuscript.

I think it also should be more clear what 'accuracy' means in a three-option outcome.

This is a valid point. Throughout the manuscript we assume that a stalemate does not contribute to accuracy, since a failed decision is not inherently right or wrong. We thus calculate accuracy as the number of trials where a correct consensus decision was made divided by the total number of trials that reached consensus. We have changed lines 80-84 to make our usage of the term 'accuracy' more comprehensive.

And finally I do not understand the sentence on ll. 100-101.

We have changed lines 107-108 for clarification (the shift in line numbers is due to the previous changes).

Again, we want to thank the referee for helping us to make the manuscript clearer and more precise and we hope that the manuscript in its current form answers the concerns raised by the referee.

Yours sincerely,

Claudia Winklmayr, Albert B. Kao, Joseph Bak-Coleman and Pawel Romanczuk